# Beyond the Permutation Symmetry of Transformers: The Role of Rotation for Model Fusion

**Binchi Zhang** [* 1]   **Zaiyi Zheng** [* 1]   **Zhengzhang Chen** [2]   **Jundong Li** [1]

## Abstract

Symmetry in the parameter space of deep neural networks (DNNs) has proven beneficial for various deep learning applications. A well-known example is the permutation symmetry in Multi-Layer Perceptrons (MLPs), where permuting the rows of weight matrices in one layer and applying the inverse permutation to adjacent layers yields a functionally equivalent model. While permutation symmetry fully characterizes the equivalence set for MLPs, its discrete nature limits its utility for transformers. In this paper, we introduce rotation symmetry, a novel form of parameter space symmetry for transformers that generalizes permutation symmetry by rotating parameter matrices in self-attention layers. Unlike permutation symmetry, rotation symmetry operates in a continuous domain, thereby significantly expanding the equivalence set for transformers. Based on this property, we propose a theoretically optimal parameter matching algorithm as a plug-and-play module to enhance model fusion. We evaluate our approach using pre-trained transformers across diverse natural language and vision tasks. Experimental results demonstrate that our rotation symmetry-based matching algorithm substantially improves model fusion, highlighting the potential of parameter space symmetry to facilitate model fusion. Our code is available on https://github.com/zhengzaiyi/RotationSymmetry.

## 1. Introduction

Parameter space symmetry is an intriguing property of neural networks that has garnered increasing attention in recent years (Du et al., 2018; Armenta & Jodoin, 2021; Kunin et al., 2021; Simsek et al., 2021; Entezari et al., 2022; Grigsby et al., 2023; Lim et al., 2024b; Ziyin et al., 2024; 2025). One of the most studied forms of parameter space symmetry is permutation symmetry (Ainsworth et al., 2023; Entezari et al., 2022). For instance, in a two-layer MLP, permuting the rows of the weight matrix in the first layer and applying the corresponding inverse permutation to the second layer results in a functionally equivalent model, *i.e.*, the outputs of the original and permuted models remain identical for any given input (Ainsworth et al., 2023). All functionally equivalent models corresponding to weight permutations form an equivalence set, which provides theoretical insights into neural network optimization, such as the linear mode connectivity of loss landscapes (Entezari et al., 2022; Zhou et al., 2023c; Ferbach et al., 2024). In addition, permutation symmetry has also proven helpful in advancing neural network applications, such as model fusion (Singh & Jaggi, 2020; Ainsworth et al., 2023) and optimization (Zhao et al., 2024; Zamir et al., 2025).

Although parameter space symmetry has been extensively studied in classical neural network architectures, such as MLPs and CNNs, the understanding of its application in transformers (Vaswani et al., 2017) remains limited. Transformers have seen rapid advancements in recent years, achieving remarkable success in a wide range of applications (Yun et al., 2019; Lewis et al., 2020; Raffel et al., 2020; Clark et al., 2020; Zhou et al., 2021; He et al., 2024; Zhu et al., 2024; Zheng et al., 2024; Zhang et al., 2025; Wei & Liu, 2025). The transformer architecture is built upon two primary submodules: feedforward networks and self-attention layers. The feedforward network, which is structurally similar to MLPs, naturally inherits the permutation symmetry that has been extensively studied in the existing literature. Self-attention layers, on the other hand, involve a unique attention mechanism powered by matrix products of queries, keys, and values, which introduce additional potentials for symmetry beyond permutations. Permutation symmetries limit the equivalence set of neural networks to *discrete operations*, which aligns well with MLPs due to their element-wise activations (*e.g.*, ReLU (Glorot et al., 2011)). In contrast, *the continuous nature* of the matrix operations in self-attention layers necessitates more flexible operations to fully characterize their equivalence set.

---
[*]Equal contribution  [1]University of Virginia  [2]NEC Laboratories America.  Correspondence to: Jundong Li <jundong@virginia.edu>.

*Proceedings of the 42nd International Conference on Machine Learning*, Vancouver, Canada. PMLR 267, 2025. Copyright 2025 by the author(s).

In this paper, we introduce **rotation symmetry**, a novel form of parameter space symmetry for self-attention layers in transformers. Specifically, we analyze the query and key matrices jointly and demonstrate that applying a rotation to the query matrix, followed by the corresponding inverse rotation to the key matrix, preserves the query-key product. Additionally, we find that the same rotation rule can be applied to the value and output matrices. Our findings provide novel insights into the functional invariance of the attention mechanism and extend the permutation symmetry (Singh & Jaggi, 2020; Wang et al., 2020; Tatro et al., 2020; Entezari et al., 2022; Ainsworth et al., 2023; Imfeld et al., 2024) from discrete spaces to continuous spaces, which significantly extends the scope of parameter space symmetries for transformers.

To further demonstrate the benefits of rotation symmetry for transformers, we explore its utility in **model fusion**. The goal of model fusion is to merge multiple well-trained end models in the parameter space to produce a single merged model with improved overall utility. Model fusion is widely adopted across various settings, such as hyperparameter tuning (where end models are trained on the same benchmark) and multi-task learning (where end models are trained with different tasks) (Jin et al., 2023). Unlike ensemble learning (Dietterich, 2000; Lakshminarayanan et al., 2017; Sagi & Rokach, 2018; Dong et al., 2020), model fusion can work in a data-agnostic manner, making it suitable for privacy-sensitive scenarios such as federated learning (Yurochkin et al., 2019; Wang et al., 2020).

The existing literature has demonstrated that the performance of model fusion is closely tied to the distance between the end models (Wortsman et al., 2022). Inspired by this finding, we propose a parameter matching algorithm that selects functionally equivalent end models from the equivalence class determined by rotation symmetry. This approach ensures that the selected representative models are closer in parameter space, resulting in a smaller inner distance. To achieve this, we formulate the problem of parameter matching as an optimization problem with orthogonal constraints. Leveraging the continuous nature of rotation symmetry, we propose a closed-form solution to this problem. Our parameter matching algorithm is highly efficient, easy to implement, and can be seamlessly incorporated as a plug-and-play module for model fusion. To evaluate its effectiveness, we conduct extensive experiments with pre-trained transformers on real-world NLP and vision tasks. The experimental results demonstrate that incorporating rotation symmetry into parameter matching improves model fusion effectively and efficiently. Furthermore, additional experiments reveal that even matching a subset of parameters can lead to notable performance improvements, highlighting the practical utility of our approach. Our contributions are threefold:

- We introduce a novel rotation symmetry for the attention mechanism in transformers, extending the concept of symmetry to a continuous space.

- Building on rotation symmetry, we propose a theoretically optimal parameter matching algorithm that improves the effectiveness of model fusion in transformers.

- Through extensive experiments, we validate the efficacy of our proposed parameter matching algorithm, demonstrating its potential to advance model fusion through parameter space symmetry.

## 2. Preliminaries

To facilitate further discussion, we begin by defining the notations. Let $\boldsymbol{A}[i,:]$, $\boldsymbol{A}[:,j]$, and $\boldsymbol{A}[i,j]$ denote the $i$-th row vector, the $j$-th column vector, and the $(i,j)$-th element of the matrix $\boldsymbol{A}$, respectively. Additionally, let $\mathbf{1}$ denote an all-ones vector. We now illustrate the concept of permutation symmetry in neural networks, using an MLP as an example. Consider an $L$-layer MLP model defined as:

$$f_{\boldsymbol{W}}(\boldsymbol{X}) = \boldsymbol{Z}^{(L)}, \ \boldsymbol{Z}^{(l)} = \sigma(\boldsymbol{Z}^{(l-1)}(\boldsymbol{W}^{(l)})^\top + \boldsymbol{b}^{(l)}), \ (1)$$

where $\boldsymbol{Z}^{(0)} = \boldsymbol{X}$ is the input feature matrix, $\boldsymbol{b}^{(l)}$ is the bias vector ($\boldsymbol{b}^{(l)}$ is a row vector and can be seen as broadcast over all rows in Equation (1)), $\boldsymbol{W} = \{\boldsymbol{W}^{(l)}, \boldsymbol{b}^{(l)}\}_{l=1,\ldots,L}$ collects all learnable parameters, and $\sigma(\cdot)$ stands for a nonlinear activation function, such as ReLU (Nair & Hinton, 2010). A loss function $\mathcal{L}(f_{\boldsymbol{W}}(\boldsymbol{X}), \boldsymbol{Y})$ is used to measure the distance between the model prediction and the ground truth label $\boldsymbol{Y}$.

To analyze the permutation symmetry in the MLP model, let $\boldsymbol{P} \in \mathcal{P}$ be a permutation matrix, where $\boldsymbol{P}[i,j] \in \{0,1\}$ and $\boldsymbol{P}[i,:]\mathbf{1} = \boldsymbol{P}[:,j]^\top\mathbf{1} = 1$ for any $i, j$. All permutation matrices are orthogonal (Strang, 1976), satisfying $\boldsymbol{P}^\top = \boldsymbol{P}^{-1}$. Consequently, for layer $l$ and $l+1$, we have:

$$\boldsymbol{Z}^{(l+1)} = \sigma\left(\sigma(\boldsymbol{Z}^{(l-1)}(\boldsymbol{W}^{(l)})^\top + \boldsymbol{b}^{(l)})(\boldsymbol{W}^{(l+1)})^\top + \boldsymbol{b}^{(l+1)}\right)$$
$$= \sigma\left(\sigma(\boldsymbol{Z}^{(l-1)}(\boldsymbol{P}^\top\boldsymbol{W}^{(l)})^\top + \boldsymbol{b}^{(l)}\boldsymbol{P})(\boldsymbol{W}^{(l+1)}\boldsymbol{P})^\top + \boldsymbol{b}^{(l+1)}\right).$$
(2)

The third equal sign holds because the element-wise activation function $\sigma$ is decoupled from column permutation (*i.e.*, being multiplied by $\boldsymbol{P}$). Based on Equation (2), it follows that for layers $l$ and $l+1$, the mappings $\boldsymbol{W}^{(l)} \to \boldsymbol{P}^\top\boldsymbol{W}^{(l)}$, $\boldsymbol{b}^{(l)} \to \boldsymbol{b}^{(l)}\boldsymbol{P}$, $\boldsymbol{W}^{(l+1)} \to \boldsymbol{W}^{(l+1)}\boldsymbol{P}$ preserves the output $\boldsymbol{Z}^{(l+1)}$ for any input $\boldsymbol{Z}^{(l-1)}$. For each pair of adjacent layers, a similar mapping exists independently based on a specific permutation matrix, denoted as $\boldsymbol{P}^{(l)}$ for layers $l$ and $l+1$. Consequently, for the entire $L$-layer MLP, there exists a mapping that preserves the model's predictions for any input $\boldsymbol{X}$: $\boldsymbol{W}^{(l)} \to (\boldsymbol{P}^{(l)})^\top\boldsymbol{W}^{(l)}\boldsymbol{P}^{(l-1)}$, $\boldsymbol{b}^{(l)} \to \boldsymbol{b}^{(l)}\boldsymbol{P}^{(l)}$. Equivalently, if we define $\boldsymbol{W}' = \{(\boldsymbol{P}^{(l)})^\top\boldsymbol{W}^{(l)}\boldsymbol{P}^{(l-1)}, \boldsymbol{b}^{(l)}\boldsymbol{P}^{(l)}\}_{l=1,\ldots,L}$ for any $\boldsymbol{P}^{(l)} \in \mathcal{P}$

$(l = 1, \ldots, L - 1)$ and $\boldsymbol{P}^{(0)} = \boldsymbol{P}^{(L)} = \boldsymbol{I}$, then we have $f_{\boldsymbol{W'}}(\boldsymbol{X}) = f_{\boldsymbol{W}}(\boldsymbol{X})$ for any $\boldsymbol{X}$. This phenomenon is referred to as the *permutation symmetry* of the parameter space (Godfrey et al., 2022; Hecht-Nielsen, 1990; Navon et al., 2023; Rossi et al., 2023; Simsek et al., 2021). Leveraging permutation symmetry allows us to identify an equivalence class of functionally equivalent model parameters, which is known as *permutation invariance* (Ainsworth et al., 2023; Entezari et al., 2022; Lubana et al., 2023). We denote the equivalence relation induced by permutation invariance as $\pi$, where, for example, $\boldsymbol{W'} = \pi(\boldsymbol{W})$ represents the equivalence between the original parameters $\boldsymbol{W}$ and the permuted parameters $\boldsymbol{W'}$.

## 3. Parameter Space Symmetry of Transformers

Transformers (Vaswani et al., 2017) have revolutionized deep learning with their ability to handle sequential data effectively, particularly in natural language processing (NLP) and other fields (Brown et al., 2020; Devlin et al., 2019; Liu et al., 2021; Radford et al., 2021). Their success is primarily driven by two key components[1]: feedforward networks and self-attention layers (Vaswani et al., 2017). To better understand the parameter space symmetry of transformers, we examine these two core modules individually in the following sections.

### 3.1. Permutation Symmetry of Feedforward Networks

We first look at the feedforward networks. The feedforward network adopted in the transformer blocks is a two-layer MLP model which can be written as

$$FFN(\boldsymbol{X}) = LN(ReLU(\boldsymbol{X}\boldsymbol{W}_i^\top + \boldsymbol{b}_i)\boldsymbol{W}_o^\top + \boldsymbol{b}_o + \boldsymbol{X}), \quad (3)$$

where $LN$ denotes the Layer Normalization operator (Ba et al., 2016). Different from (Vaswani et al., 2017), we include the residual connection (He et al., 2016) and layer normalization modules into the formula of the feedforward network (and the self-attention layer mentioned later). According to Equation (2) and the analysis in Preliminaries, the feedforward networks have the permutation symmetry property and the equivalence class determined by permutation invariance is defined as

$$\boldsymbol{W}_i \rightarrow \boldsymbol{P}^\top \boldsymbol{W}_i, \; \boldsymbol{b}_i \rightarrow \boldsymbol{b}_i \boldsymbol{P}, \; \boldsymbol{W}_o \rightarrow \boldsymbol{W}_o \boldsymbol{P}, \; \boldsymbol{b}_o \rightarrow \boldsymbol{b}_o, \quad (4)$$

where $\boldsymbol{P} \in \mathcal{P}$ is a permutation matrix. It is worth noting that the permutations of different feedforward networks in a transformer are independent due to the scalable modular design. Consequently, we are able to flexibly compute the permutation invariance equivalence class of each $FFN$ module in a transformer model.

---

[1]We follow the architecture in the original transformer paper (Vaswani et al., 2017).

### 3.2. Rotation Symmetry of Self-attention Layers

We then focus on the self-attention layers. In this paper, we introduce the *rotation symmetry* of self-attention layers. For MLPs, we have to switch the order of multiplying $\boldsymbol{P}$ and passing the activation function $\sigma$ (the third equal sign in Equation (2)), requiring the matrix $\boldsymbol{P}$ to be a permutation matrix. In contrast, the self-attention layer does not contain an element-wise activation function, enabling a wider range of the matrix $\boldsymbol{P}$ in self-attention layers. The self-attention layer can be written as

$$ATTN(\boldsymbol{X}) = LN\left(Cat_{h=1}^H \left\{\boldsymbol{X}_{QKV}^h\right\} \boldsymbol{W}_O^\top + \boldsymbol{b}_O + \boldsymbol{X}\right),$$
$$\boldsymbol{X}_{QKV}^h = Sftmx\left(\boldsymbol{X}_Q^h(\boldsymbol{X}_K^h)^\top / \sqrt{d_k}\right) \cdot \boldsymbol{X}_V^h,$$

where $Cat$ stands for the operator concatenating the outputs of multi-head attention, $Sftmx(\cdot)$ denotes the softmax operator, $H$ denotes the number of multi-heads, and the subscripts $Q$, $K$, $V$, and $O$ denote Query, Key, Value, and Output, respectively.

We first transform the query and key matrices as

$$\begin{aligned}\boldsymbol{X}_Q^h(\boldsymbol{X}_K^h)^\top &= (\boldsymbol{X}(\boldsymbol{W}_Q^h)^\top + \boldsymbol{b}_Q^h)(\boldsymbol{X}(\boldsymbol{W}_K^h)^\top + \boldsymbol{b}_K^h)^\top \\ &= (\boldsymbol{X}(\boldsymbol{W}_Q^h)^\top + \boldsymbol{b}_Q^h)\boldsymbol{R}\boldsymbol{R}^\top(\boldsymbol{X}(\boldsymbol{W}_K^h)^\top + \boldsymbol{b}_K^h)^\top \\ &= (\boldsymbol{X}(\boldsymbol{R}^\top \boldsymbol{W}_Q^h)^\top + \boldsymbol{b}_Q^h \boldsymbol{R})(\boldsymbol{X}(\boldsymbol{R}^\top \boldsymbol{W}_K^h)^\top + \boldsymbol{b}_K^h \boldsymbol{R})^\top,\end{aligned}$$

where $\boldsymbol{R}$ is a rotation matrix, *i.e.*, $\boldsymbol{R}\boldsymbol{R}^\top = \boldsymbol{I}$. It is worth noting that each multi-head corresponds to a specific rotation matrix $\boldsymbol{R}$. Let $\boldsymbol{W}_O = [\boldsymbol{W}_O^1 \; \boldsymbol{W}_O^2 \; \cdots \; \boldsymbol{W}_O^H]$ and we then rewrite the concatenating operation (with the product by $\boldsymbol{W}_O$) as $\sum_{h=1}^H S(\cdots)(\boldsymbol{X}(\boldsymbol{W}_V^h)^\top + \boldsymbol{b}_V^h)(\boldsymbol{W}_O^h)^\top$. Similar to the Q-K case, we can transform the value and output matrices as $(\boldsymbol{X}(\boldsymbol{W}_V^h)^\top + \boldsymbol{b}_V^h)(\boldsymbol{W}_O^h)^\top = (\boldsymbol{X}(\boldsymbol{W}_V^h)^\top + \boldsymbol{b}_V^h)\boldsymbol{R}\boldsymbol{R}^\top(\boldsymbol{W}_O^h)^\top = (\boldsymbol{X}(\boldsymbol{R}^\top \boldsymbol{W}_V^h)^\top + \boldsymbol{b}_V^h \boldsymbol{R})(\boldsymbol{W}_O^h \boldsymbol{R})^\top$ for each multi-head $h$, where $\boldsymbol{R}$ is a rotation matrix. Finally, we derive an equivalence class of the parameters in a self-attention layer determined by rotation invariance as

$$\begin{aligned}&\boldsymbol{W}_Q^h \rightarrow (\boldsymbol{R}_{qk}^h)^\top \boldsymbol{W}_Q^h, \; \boldsymbol{b}_Q^h \rightarrow \boldsymbol{b}_Q^h \boldsymbol{R}_{qk}^h, \\ &\boldsymbol{W}_K^h \rightarrow (\boldsymbol{R}_{qk}^h)^\top \boldsymbol{W}_K^h, \; \boldsymbol{b}_K^h \rightarrow \boldsymbol{b}_K^h \boldsymbol{R}_{qk}^h, \boldsymbol{W}_V^h \rightarrow (\boldsymbol{R}_{vo}^h)^\top \boldsymbol{W}_V^h, \\ &\boldsymbol{b}_V^h \rightarrow \boldsymbol{b}_V^h \boldsymbol{R}_{vo}^h, \; \boldsymbol{W}_O^h \rightarrow \boldsymbol{W}_O^h \boldsymbol{R}_{vo}^h, \; \boldsymbol{b}_O \rightarrow \boldsymbol{b}_O,\end{aligned}$$
$$(5)$$

where $\boldsymbol{R}_{qk}^h$ and $\boldsymbol{R}_{vo}^h$ are rotation matrices for $h = 1, \ldots, H$. The progress from permutation to rotation extends the symmetry of transformers to a continuous space and enhances our understanding of the parameter space symmetry of attention mechanism. The denseness of the symmetry allows for the choice of better invariant models to analyze transformers' loss landscapes. For better clarity, we provide an intuitive illustration of the rotation symmetry in Figure 1.

## 4. Symmetry for Model Fusion

In this section, we explore the benefit of the rotation symmetry of transformers in model fusion (Li et al., 2023; Matena

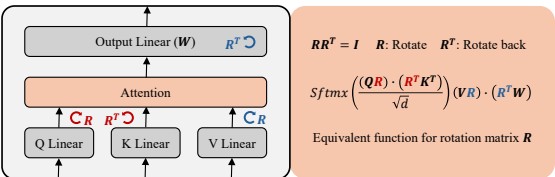

*Figure 1.* The rotation symmetry of self-attention layers.

& Raffel, 2022; Wortsman et al., 2022; Yadav et al., 2023; Jin et al., 2023; Daheim et al., 2024; Yang et al., 2024). Model fusion is proposed to merge multiple given end models trained in different settings (*e.g.*, upon different datasets and hyperparameter settings) in the parameter space to improve model utility and robustness. Compared with ensemble learning (Dietterich, 2000; Sagi & Rokach, 2018; Lakshminarayanan et al., 2017), model fusion has a lower inference-stage complexity without requiring access to the training data. Most existing methods of model fusion conduct a weighted averaging of different end models, *e.g.*, direct averaging (Wortsman et al., 2022), Fisher-weighted averaging (Matena & Raffel, 2022), and regression-mean averaging (Jin et al., 2023). We next show the potential of exploiting the permutation and rotation symmetry as a plug-and-play module to improve the model fusion techniques.

### 4.1. Background and Motivation

Let $\boldsymbol{W}_1, \ldots, \boldsymbol{W}_k$ denote $k$ different end models (after training or pre-training) with the same architecture. The goal of model fusion is to merge the given $k$ end models in the parameter space and obtain a single model. If the given models are trained over different datasets, we can expect the merged model to have better utility and out-of-distribution robustness (Jin et al., 2023). The theoretical results in previous literature (Wortsman et al., 2022) have shown that *strong convexity* and *closer end models* can boost the utility of direct model fusion. Consequently, the primary advantage of permutation (and rotation) symmetry applying to model fusion is that we can substitute the end models with corresponding carefully chosen equivalent models (in the equivalence class determined by permutation and rotation invariance) to make the selected end models more concentrated, *i.e.*, closer to each other. This step is usually called **parameter matching** (Singh & Jaggi, 2020; Wang et al., 2020; Ainsworth et al., 2023), aka parameter or neuron alignment. Parameter matching, which aims to reduce the distance between end models, can naturally yield closer end models and improve model fusion performance. On the other hand, different end models can lie in the basins of different local optimums regarding the highly non-convex nature of transformers. Previous studies have verified that parameter matching can merge different end models toward a single low-loss basin, *i.e.*, the loss value along the linear interpolation between matched models shows an approxi-

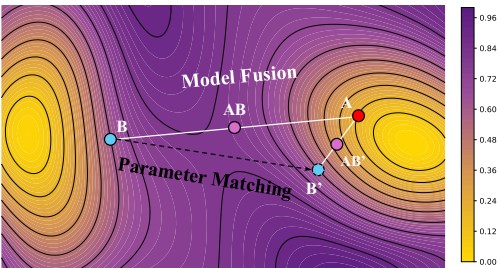

*Figure 2.* An intuitive example of the usage of parameter space symmetry for model fusion. The background shows the contour map of the loss landscape in the model parameter space. A and B are the original end models to be merged, AB is the result of naive model fusion, and AB' is the result of model fusion with parameter matching.

mately flat or convex curve (Entezari et al., 2022; Ainsworth et al., 2023). This property is called *linear mode connectivity* and is regarded as a weak form of convexity (Ainsworth et al., 2023). As a result, parameter matching can also help improve the convexity of the objective in the area adjacent to the end models. We showcase an intuitive example of using parameter space symmetry to improve model fusion in Figure 2 to illustrate the intuition behind parameter matching. We can observe that the merged model after parameter matching (AB') has a lower loss value, *i.e.*, better utility than the naive merged model (AB).

### 4.2. Parameter Matching

Next, our primary goal is to develop a practical parameter matching algorithm to minimize the distance between different end models. We begin by merging two models $\boldsymbol{W}_1$ and $\boldsymbol{W}_2$.

**Matching two FFNs.** Let $\{\boldsymbol{W}_{i_k}, \boldsymbol{b}_{i_k}, \boldsymbol{W}_{o_k}, \boldsymbol{b}_{o_k}\}_{k=1,2}$ denote the model parameters in the two feedforward networks to be merged where $k$ denote the index of the networks. We have already derived the equivalence relation of parameters in the feedforward networks determined by permutation invariance as Equation (4). Based on Equation (4), we can formulate parameter matching as an optimization problem as follows.

$$\operatorname{argmin}_{\boldsymbol{P}_1, \boldsymbol{P}_2 \in \mathcal{S}} \| \boldsymbol{P}_1^\top \boldsymbol{W}_{i_1} - \boldsymbol{P}_2^\top \boldsymbol{W}_{i_2} \|_F^2$$
$$+ \| \boldsymbol{b}_{i_1} \boldsymbol{P}_1 - \boldsymbol{b}_{i_2} \boldsymbol{P}_2 \|_F^2 + \| \boldsymbol{W}_{o_1} \boldsymbol{P}_1 - \boldsymbol{W}_{o_2} \boldsymbol{P}_2 \|_F^2. \quad (6)$$

As shown in Equation (6), the goal is to find the functionally equivalent models from the equivalence classes $\pi(\boldsymbol{W}_1)$ and $\pi(\boldsymbol{W}_2)$ determined by $\boldsymbol{P}_1$ and $\boldsymbol{P}_2$, which has the smallest $\ell$-2 distance in the parameter space. Following the method (Ainsworth et al., 2023), we reformulate the optimization problem shown in Equation (6) as a *linear assignment problem*:

$$\operatorname{argmax}_{\boldsymbol{P} \in \mathcal{S}} \left\langle \boldsymbol{P}, \boldsymbol{W}_{i_1} \boldsymbol{W}_{i_2}^\top + \boldsymbol{b}_{i_1}^\top \boldsymbol{b}_{i_2} + \boldsymbol{W}_{o_1}^\top \boldsymbol{W}_{o_2} \right\rangle_F. \quad (7)$$

Linear assignment problems such as Equation (7) have been well-studied in previous literature (Martello & Toth, 1987; Burkard & Cela, 1999) and can be solved precisely by Hungarian Algorithm (Martello & Toth, 1987). After calculating the value of $P$, we substitute $P = P_1 P_2^{-1}$ back to Equation (4) and let $P_2 = I$ (for simplicity), then we obtain the matched parameter as $W_{i_1} \rightarrow P^\top W_{i_1}$, $b_{i_1} \rightarrow b_{i_1} P$, $W_{o_1} \rightarrow W_{o_1} P$, $b_{o_1} \rightarrow b_{o_1}$. Parameters of the second model $\{W_{i_2}, b_{i_2}, W_{o_2}, b_{o_2}\}$ remain unchanged, which makes the second model an anchor model for matching.

**Matching two ATTNs.** Let the parameters in the two self-attention layers to be merged be $\{W_{Q_k}^i, b_{Q_k}^i, W_{K_k}^i, b_{K_k}^i, W_{V_k}^i, b_{V_k}^i, W_{O_k}^i, b_{O_k}^i\}_{k=1,2;i=1\ldots H}$ where $k$ denote the index of the layers. The equivalence relation of these parameters in the self-attention layers in terms of rotation invariance is shown in Equation (5). Similar to matching FFNs, our goal is to match the parameters of one (source) attention layer to the other (target) attention layer where the matched parameters are supposed to be *closest* to the target parameters while being functionally equivalent when feeding with any input data, *i.e.*, in the equivalence class determined by rotation invariance. The goal can be formulated as an optimization problem.

$$
\begin{aligned}
\min_{R_{qk}^h, R_{vo}^h \in \mathcal{R}} & \left\| \begin{bmatrix} (W_{Q_1}^h)^\top & (W_{Q_2}^h)^\top \\ b_{Q_1}^h & b_{Q_2}^h \end{bmatrix} \begin{bmatrix} R_{qk1}^h \\ -R_{qk2}^h \end{bmatrix} \right\|_F^2 \\
& + \left\| \begin{bmatrix} (W_{K_1}^h)^\top & (W_{K_2}^h)^\top \\ b_{K_1}^h & b_{K_2}^h \end{bmatrix} \begin{bmatrix} R_{qk1}^h \\ -R_{qk2}^h \end{bmatrix} \right\|_F^2 \\
& + \left\| \begin{bmatrix} (W_{V_1}^h)^\top & (W_{V_2}^h)^\top \\ b_{V_1}^h & b_{V_2}^h \end{bmatrix} \begin{bmatrix} R_{vo1}^h \\ -R_{vo2}^h \end{bmatrix} \right\|_F^2 \\
& + \left\| \begin{bmatrix} W_{O_1}^h & W_{O_2}^h \\ 0 & 0 \end{bmatrix} \begin{bmatrix} R_{vo1}^h \\ -R_{vo2}^h \end{bmatrix} \right\|_F^2,
\end{aligned} \tag{8}
$$

where $\mathcal{R}$ denotes the set of rotation matrices. To solve this problem, we divide Equation (8) into two separate optimization problems (the first line in terms of $R_{qk1}^h$, $R_{qk2}^h$ as the first objective and the second line in terms of $R_{vo1}^h$, $R_{vo2}^h$ as the second objective). Considering the similar formulation of these two optimization problems, in this paper, we propose the following theorem to solve both problems.

**Theorem 4.1.** *The following optimization problem has a closed-form solution.*

$$
\begin{aligned}
\min_{R_1, R_2 \in \mathcal{R}} & \left\| \begin{bmatrix} W_{Q_1}^\top & W_{Q_2}^\top \\ b_{Q_1} & b_{Q_2} \end{bmatrix} \begin{bmatrix} R_1 \\ -R_2 \end{bmatrix} \right\|_F^2 \\
& + \left\| \begin{bmatrix} W_{K_1}^\top & W_{K_2}^\top \\ b_{K_1} & b_{K_2} \end{bmatrix} \begin{bmatrix} R_1 \\ -R_2 \end{bmatrix} \right\|_F^2.
\end{aligned} \tag{9}
$$

*The solution is given by*

$$
R_1 = UV^\top, \quad R_2 = I, \tag{10}
$$

**Algorithm 1** Matching two self-attention layers.

**Input:** Model parameters of the attention layers $\{W_{Q_k}^h, b_{Q_k}^h, W_{K_k}^h, b_{K_k}^h, W_{V_k}^h, b_{V_k}^h, W_{O_k}^h, b_{O_k}^h\}_{k=1,2;h=1,\ldots,H}$.
**Output:** The optimally matched parameters from source parameters ($k = 1$) to anchor parameters ($k = 2$, maintain unchanged).

1: QK solution: $R_{qk1}^h = U_{qk}^h (V_{qk}^h)^\top$, $R_{qk2}^h = I$, where $U_{qk}^h \Sigma_{qk}^h (V_{qk}^h)^\top = W_{Q_1}^h (W_{Q_2}^h)^\top + W_{K_1}^h (W_{K_2}^h)^\top + (b_{Q_1}^h)^\top b_{Q_2}^h + (b_{K_1}^h)^\top b_{K_2}^h$.
2: QK matching: $W_{Q_1}^h \rightarrow (R_{qk1}^h)^\top W_{Q_1}^h$, $W_{K_1}^h \rightarrow (R_{qk1}^h)^\top W_{K_1}^h$, $b_{Q_1}^h \rightarrow b_{Q_1}^h R_{qk1}^h$, $b_{K_1}^h \rightarrow b_{K_1}^h R_{qk1}^h$.
3: VO solution: $R_{vo1}^h = U_{vo}^h (V_{vo}^h)^\top$, $R_{vo2}^h = I$, where $U_{vo}^h \Sigma_{vo}^h (V_{vo}^h)^\top = W_{V_1}^h (W_{V_2}^h)^\top + (W_{O_1}^h)^\top W_{O_2}^h + (b_{V_1}^h)^\top b_{V_2}^h$.
4: VO matching: $W_{V_1}^h \rightarrow (R_{vo1}^h)^\top W_{V_1}^h$, $W_{O_1}^h \rightarrow W_{O_1}^h R_{vo1}^h$, $b_{V_1}^h \rightarrow b_{V_1}^h R_{vo1}^h$.

where $I$ is the identity matrix and $U\Sigma V^\top = W_{Q_1} W_{Q_2}^\top + W_{K_1} W_{K_2}^\top + b_{Q_1}^\top b_{Q_2} + b_{K_1}^\top b_{K_2}$ is the result of eigendecomposition.

We leave a detailed proof of Theorem 4.1 in Appendix A. According to Theorem 4.1, we can obtain the algorithm to match two self-attention layers shown in Algorithm 1. Algorithm 1 can be seen as an adaptation of the Kabsch algorithm (Kabsch, 1976; Umeyama, 1991). Without loss of generality, we let the parameters in the second ($k = 2$) self-attention layer be the anchor and conduct rotation for the first layer ($k = 1$). The denseness of rotation symmetry helps reduce the distance between the end models after parameter matching as Algorithm 1 shows.

**Rescaling Matching.** We find that the rescaling symmetry (Neyshabur et al., 2015; Meng et al., 2019; Godfrey et al., 2022; Kalogeropoulos et al., 2024) can be integrated with our proposed rotation symmetry in the parameter matching algorithm. For instance, for the (simplified) Q-K product $W_Q W_K$, we can find a rescaling operation that preserves the functionality of the Q-K product $aW_Q \cdot \frac{1}{a} W_K$ where $a \neq 0$ is a real number. By adding a scalar variable to each parameter matrix, we can formulate the objective of rescaling symmetry as follows (taking the Q-K product as an example).

$$
\begin{aligned}
\min_a & \left\| aW_{Q_1}' - W_{Q_2}' \right\|^2 + \left\| ab_{Q_1}' - b_{Q_2}' \right\|^2 \\
& + \left\| W_{K_1}'/a - W_{K_2}' \right\|^2 + \left\| b_{K_1}'/a - b_{K_2}' \right\|^2,
\end{aligned} \tag{11}
$$

where $a$ is the rescaling variable and $'$ denotes the parameters after rotation symmetry-based matching. Here, we still set model 2 as the anchor model and conduct the rescaling operation on model 1 to align with model 2. We can easily

*Table 1.* Experimental results of in-domain (Emotion and NER) and out-of-domain (NER-CoNLL) model fusion for two base language models over Emotion and NER tasks.

| | Emotion | | NER | | NER-CoNLL | |
|---|---|---|---|---|---|---|
| | Roberta | Deberta | Roberta | Deberta | Roberta | Deberta |
| Simple | 35.87 | 2.99 | 60.88 | 27.54 | 26.86 | 10.80 |
| +match | 35.87 | 2.99 | 60.88 | **31.31** | **26.87** | **23.30** |
| Fisher | 44.02 | 35.95 | 54.55 | 33.20 | **23.06** | 12.53 |
| +match | **44.05** | **35.98** | **54.58** | **33.83** | 23.05 | **18.33** |
| Regmean | 35.87 | 2.99 | 60.88 | 27.54 | 26.86 | 10.80 |
| +match | **39.95** | 2.99 | 60.88 | **31.31** | **26.87** | **14.06** |

*Table 2.* Experimental results of ViT merging over the image classification task.

| | w/o match | w/o rescale | match |
|---|---|---|---|
| Simple | 7.60 | 10.22 | 10.19 |
| Fisher | 17.96 | 18.61 | 18.58 |
| Regmean | 14.24 | 15.31 | 15.35 |
| OT-ACTS-EMD | 32.08 | 32.50 | 32.53 |
| OT-ACTS | 61.15 | 61.23 | 61.25 |
| OT-WTS | 57.11 | 57.16 | 57.17 |

solve the optimality condition of Equation (11) as

$$
\left(\|\boldsymbol{W}'_{Q_1}\|^2 + \|\boldsymbol{b}'_{Q_1}\|^2\right) a^4 - \|\boldsymbol{W}'_{K_1}\|^2 - \|\boldsymbol{b}'_{K_1}\|^2
$$
$$
- \left(\langle \boldsymbol{W}'_{Q_1}, \boldsymbol{W}'_{Q_2}\rangle + \langle \boldsymbol{b}'_{Q_1}, \boldsymbol{b}'_{Q_2}\rangle\right) a^3
$$
$$
+ \left(\langle \boldsymbol{W}'_{K_1}, \boldsymbol{W}'_{K_2}\rangle + \langle \boldsymbol{b}'_{K_1}, \boldsymbol{b}'_{K_2}\rangle\right) a = 0.
\tag{12}
$$

The roots of Equation (12) can be derived using numerical methods as the value of $a$. We then conduct the rescaling operation to model 1 as $\boldsymbol{W}'_{Q_1} \to a\boldsymbol{W}'_{Q_1}$, $\boldsymbol{b}'_{Q_1} \to a\boldsymbol{b}'_{Q_1}$, $\boldsymbol{W}'_{K_1} \to \frac{1}{a}\boldsymbol{W}'_{K_1}$, $\boldsymbol{b}'_{K_1} \to \frac{1}{a}\boldsymbol{b}'_{K_1}$. It is worth noting that the rescaling symmetry-based matching is conducted after the rotation symmetry-based matching. Using the rescaling operation, we extend the rotation matrices to orthogonal matrices with larger norms. Nevertheless, the end models are close to each other in practical scenarios so the value of $a$ is usually close to 1.

**Complexity.** We next provide a brief analysis of the complexity of our proposed parameter matching algorithm. We let the hidden dimension of the target transformer be $d$, the layer of the target transformer be $L$, and the number of attention heads be $H$. For the parameter matching of feedforward networks, the linear assignment problem can be solved in $O(d^3)$ by the Hungarian algorithm (Kuhn, 1955; Martello & Toth, 1987). Additionally, to solve the optimization problem in Equation (8), Algorithm 1 requires the complexity of $O(d^3)$ for eigendecomposition. Hence, the complexity of our proposed full parameter matching algorithm of a transformer is $O(d^3 LH)$, similar to the complexity of the feedforward. It is worth noting that the complexity can be further reduced in two ways. The first way is to match a subset of layers instead of all. In this way, the complexity of our proposed parameter matching algorithm is $O(d^3 L_s H)$ where $L_s$ denotes the number of selected layers. The second way is to match each unit module (a single feedforward network or a single attention layer) in parallel. The decoupling of matching different modules makes it easy to implement multiprocessing. Consequently, the overall complexity becomes to $O(d^3 LH/p)$ where $p$ is the number of processes in parallel.

## 5. Experiments

In this section, we conduct experiments to evaluate the effectiveness of our proposed parameter matching algorithm based on rotation symmetry. Specifically, we aim to answer the following research questions: RQ1: Can our proposed parameter matching algorithm enhance the performance of model fusion for transformer-based models? RQ2: How does rotation symmetry contribute to parameter matching for self-attention layers? RQ3: How does parameter matching influence the loss landscape between independently trained models? RQ4: As a plugin module, does our algorithm introduce significant additional computational overhead? RQ5: Is matching all transformer layers equally important? Can we improve efficiency without compromising utility by matching only a subset of layers?

### 5.1. Experimental Settings

**Platform.** Our implementation is based on `Python 3.10` and `Pytorch 1.13`. All fine-tuning, parameter matching, and model merging processes are conducted on a cluster equipped with `Nvidia A100 80GB` GPUs.

**Models.** To evaluate the effectiveness of our approach, we use two widely adopted transformer models: RoBERTa (Liu et al., 2019) and DeBERTa (He et al., 2021). We obtain pretrained models of RoBERTa (`roberta-base` with 12 attention layers) and DeBERTa-Large (`microsoft/deberta-v3-large` with 24 attention layers) from the Hugging Face library. For vision transformers (ViTs) (Dosovitskiy et al., 2021), we directly use the pretrained models following (Imfeld et al., 2024). By selecting these three type of models, we assess the effectiveness of our method across different model scales and downstream tasks.

**Finetuning and Matching.** In the experiments, each pretrained language model is fine-tuned for 20 epochs on each dataset individually. We set the learning rate at 1e-5, the batch size at 16, and the warmup ratio at 0.06 for each model. After fine-tuning, we perform parameter matching and model merging, considering both in-domain (pairwise

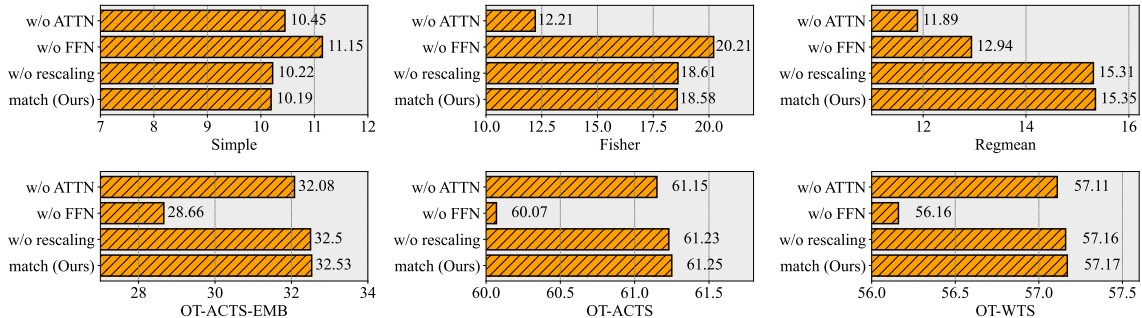

*Figure 3.* Ablation Study of ViT merging over the image classification task. "ATTN" is short for "attention". We compare our matching algorithm with its three variants (w/o ATTN/FFN/rescaling) and the original performance (w/o match) on all six merging baselines.

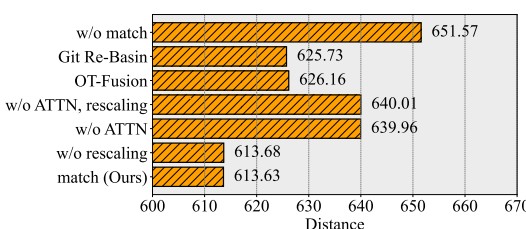

*Figure 4.* The Euclidean Distance of end ViT models after different parameter matching algorithms.

fine-tuned models) and out-of-domain (grouped models) experiments. Notably, we match only the parameters within the attention layers, while the classifier module is directly copied from the model fine-tuned on the corresponding downstream task. For additional details on datasets and baseline methods, please refer to Appendix C.

### 5.2. Performance of Model Fusion

To answer RQ1, we compare the performance of three model fusion baselines with and without our proposed parameter matching algorithm.

**In-Domain Settings.** We evaluate model fusion performance on Emotion Classification and Named Entity Recognition (NER) tasks, where our approach (**+match**) is integrated as a plugin module for three model fusion baselines. For each task, we fine-tune the language models on each in-domain dataset (5 for Emotion, 6 for NER) respectively and merge them pairwise. The first two columns in Table 1 present the average macro F1-score (for Emotion) and micro F1-score (for NER) of the merged models following (Jin et al., 2023).

**Out-of-Domain Settings.** To assess the generalization ability, we merge models trained on all in-domain NER datasets and evaluate their performance on CoNLL datasets, which serve as out-of-domain (OOD) test sets. The third column in Table 1 reports the OOD performance of the merged models.

**ViT Settings.** We employ three OT (optimal transportation)-based methods (`OT-ACTS-EMD`, `OT-ACTS`, and `OT-WTS`) from (Imfeld et al., 2024) as additional baselines. Two pretrained models are merged, and we evaluate their performance on the CIFAR-10 (Krizhevsky et al., 2009) dataset. Table 2 presents the classification accuracy of the merged models. To improve the flexibility of parameter matching, we match each layer separately before conducting model fusion. The merged model achieving the highest validation performance is selected as the final test model.

From the results in Table 1 and Table 2, we make the following observations: (1) Our parameter matching algorithm consistently improves the performance of different model fusion methods. The reason is that parameter matching helps align distant models, bringing them closer in parameter space. As a result, the merged model is more likely to approach an overall minimum. (2) Compared to RoBERTa-base, our method brings larger improvement for DeBERTa-large, suggesting that larger models benefit more from parameter matching. We explain this as larger models have more parameters that can be aligned, and in high-dimensional spaces, models tend to be more spread out. Subsequently, parameter matching helps bridge this gap more effectively. (3) Among all the fusion methods, simple fusion shows the largest improvement after parameter matching. This is likely because simple fusion directly averages model weights, so reducing the distance between model parameters brings a clear benefit. In contrast, advanced fusion methods work by aligning outputs of the end models based on input data, which is equivalent to weighted averaging. Unlike direct averaging, these methods benefit from parameter matching in a more implicit way.

### 5.3. Ablation Study

To answer RQ2, we conduct ablation studies to evaluate the contributions of rotation symmetry in matching different model components. These experiments are performed on the same pair of pretrained ViT models used in the previous

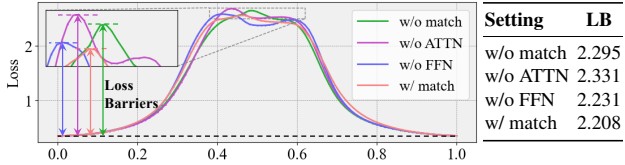

Figure 5. Loss landscapes and barriers between the two pretrained ViT models under four distinct matching settings. "LB" is short for "Loss Barrier".

Table 3. The average runtime in seconds of the fine-tuning (top), matching (middle), and merging (bottom) stage.

|                 | Deberta  | Roberta | ViT   |
|-----------------|----------|---------|-------|
| Fine-tuning     | 12983.71 | 3071.49 | -     |
| Matching        | 1.59     | 1.71    | 3.48  |
| Simple Merging  | 0.13     | 0.09    | 0.22  |
| Fisher Merging  | 197.47   | 69.57   | 83.67 |
| Regmean Merging | 137.67   | 36.44   | 71.02 |

section. We design three ablation settings: (1) *w/o ATTN*: Matches only the FFN modules, excluding self-attention modules. (2) *w/o FFN*: Matches only the self-attention modules, excluding FFN modules. (3) *w/o rescaling*: Disables the rescaling symmetry in our algorithm. Ablation results in Figure 3 demonstrate the following observations: (1) In Fisher and RegMean, attention matching contributes more significantly to the fusion process. (2) In contrast, OTFusion benefits more from FFN matching. These distinctions demonstrate the nuanced interactions between the fusion mechanisms and corresponding component alignment strategies. (3) *w/o rescaling* results in performance degradation across most settings, validating the effectiveness of rescaling symmetry integrated with rotation symmetry.

To further quantify the influence of rotation symmetry, we measure the Euclidean distance between the matched model and the anchor model in the parameter space. Instead of aggregating distances at the layer or module level, we analyze the entire model holistically to provide a more comprehensive assessment of alignment. We also compare our method against two external baselines: Git Re-Basin (Ainsworth et al., 2023) and OT-Fusion (Imfeld et al., 2024). Following the original paper, OT-Fusion is applied only to FFN modules. The results in Figure 4 indicate that our rotation symmetry-based parameter matching algorithm consistently reduces the distance between end models more effectively than permutation symmetry-based methods. This improvement can be attributed to the continuous nature of rotation symmetry, which allows for smoother and more precise alignment, especially in self-attention layers. Furthermore, the integration of rescaling symmetry leads to additional distance reductions of end models, further enhancing parameter alignment.

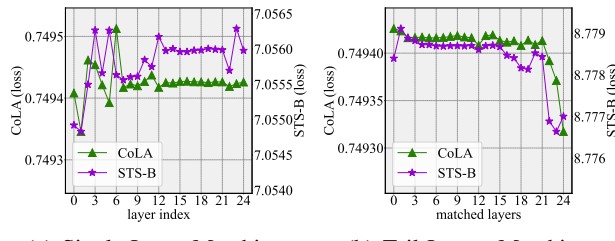

(a) Single-Layer Matching    (b) Tail-Layers Matching

Figure 6. Evaluation loss of matching a subset of layers.

### 5.4. Loss Landscape Study

To address RQ3, we examine the loss landscape between the two pretrained ViT models introduced in Section 5.2, both with and without parameter matching. Specifically, we first analyze the cross entropy loss values $\mathcal{L}(\theta(\alpha))$ along the linear interpolation path between the end model parameters $\theta(\alpha) = \alpha \cdot \theta_A + (1-\alpha) \cdot \theta_B, \quad \alpha \in [0, 1]$, where $\theta_A$ and $\theta_B$ denote the end models. Next, we separately apply attention-only matching (*w/o FFN*), FFN-only matching (*w/o ATTN*), and complete matching (*w/ match*) to $\theta_B$ before constructing the same linear interpolation path with respect to $\theta_A$. The results in Figure 5 give rise to the following observations: (1) All model pairs exhibit positive loss barriers, i.e., the range of loss value along the interpolation path, highlighting the strong inherent non-convexity of ViTs (Park & Kim, 2022). (2) Complete matching yields the lowest loss barrier, indicating that aligning parameters through rotation symmetry leads to smoother connectivity between model pairs. (3) While FFN-only matching (*w/o ATTN*) has been proven effective for MLPs and CNNs (Ainsworth et al., 2023), its effectiveness remains limited for Transformers. Their intricate loss landscape brings additional challenges for parameter matching.

### 5.5. Complexity Study

To answer RQ4, we evaluate the computational overhead introduced by our parameter matching algorithm. Specifically, we measure the average runtime for fine-tuning (per dataset), matching (per model pair), and merging methods (per model pair) from the main experiments, as shown in Table 3. For ViTs, we only use pre-trained models following (Imfeld et al., 2024) without fine-tuning. The results show that our parameter matching module incurs an overhead of less than 5% compared to Fisher / RegMean merging across all models, with negligible impact on overall complexity.

### 5.6. Matching a Subset of Layers

To answer RQ5, we investigate the effect of matching different subsets of layers. We fine-tune DeBERTa models on CoLA and STS-B from the GLUE benchmark and evaluate the performance of matched models under different subsets

of layers. Specifically, we cross-validate the importance of each attention layer in matching through two settings:

**Single-Layer Matching.** In this setting, we match only a single attention layer while leaving all other layers unchanged. The evaluation loss corresponding to different matched layer indices is shown in Figure 6(a).

**Tail-Layers Matching.** Here, we match a certain number of trailing attention layers (*e.g.*, when matching three attention layers, we align only the last three layers). The evaluation loss for different numbers of matched layers is presented in Figure 6(b).

The experimental results in Figure 6 demonstrate that matching head layers yields greater improvements in model utility compared to tail layers. Notably, the loss value drops sharply when the first 5 layers are matched. Based on this observation, we can further improve efficiency by dropping tail layers without significantly compromising utility.

# 6. Related Work

**Parameter Space Symmetry.** Parameter space symmetry refers to a set of models with different parameter values but functionally equivalent. This concept has been extensively studied in the context of deep neural networks, as it plays a crucial role in understanding model behavior and training dynamics. Examples of parameter space symmetries include rescaling symmetry (Neyshabur et al., 2015; Badrinarayanan et al., 2015; Du et al., 2018; Meng et al., 2019), scaling symmetry (Kunin et al., 2021), and translation symmetry (Kunin et al., 2021). These symmetries have been identified in conventional deep neural networks to provide deeper insights into training dynamics and to accelerate the optimization process (Zhao et al., 2022; 2023; 2024). Another important type of parameter space symmetry is permutation symmetry, which has been shown to closely relate to the manifold of global minima and critical points (Fukumizu & Amari, 2000; Brea et al., 2019; Simsek et al., 2021; Benton et al., 2021; Entezari et al., 2022; Ainsworth et al., 2023). The permutation symmetry can also be used to align (match) the outputs or model parameters of different end models with the same architecture (Singh & Jaggi, 2020; Wang et al., 2020; Ainsworth et al., 2023; Peña et al., 2023; Imfeld et al., 2024; Navon et al., 2024). Complementary to parameter space symmetry, recent advances in neural functionals and metanetworks (Navon et al., 2023; Zhou et al., 2023b;a; Lim et al., 2024a; Tran et al., 2024) explore permutation-equivariant functionals that operate directly on model weights across diverse architectures. Some concurrent works (Liu, 2024; Tran et al., 2024) investigate similar forms of rotation symmetry in neural networks. Liu (2024) shows that the mirror symmetry leads

to low-rankness. Meanwhile, Tran et al. (2024) leverages the rotation symmetry to construct transformer-based neural functional networks. In comparison, our study focuses on the role of rotation symmetry in model fusion and proposes a theoretically optimal parameter matching approach based on the properties of rotation symmetries.

**Model Fusion.** The goal of model fusion (Li et al., 2023) is to merge multiple available end models (with the same architecture) to obtain a stronger model. The scenarios of model fusion can be flexible. When training on the same dataset, model fusion can be used to improve the model utility or generalization by merging models trained with different configurations or in different stages (Izmailov et al., 2018; Gupta et al., 2020; Cha et al., 2021; Wortsman et al., 2022; Rame et al., 2022; Arpit et al., 2022; Huang et al., 2024; Yadav et al., 2024; Hammoud et al., 2024). As a representative method in this setting, ModelSoup (Wortsman et al., 2022) greedily averages the models fine-tuned with different hyperparameter configurations to improve the utility and robustness of the model. In addition, when training on different datasets or tasks, model fusion can be used to improve out-of-domain generalization or multitasking of the model (Matena & Raffel, 2022; Choshen et al., 2022; Li et al., 2022; Jin et al., 2023; Zhou et al., 2024), especially for language models. A state-of-the-art merging algorithm, RegMean (Jin et al., 2023), successfully merges language models fine-tuned over different tasks and improves the model's out-of-distribution generalization. Moreover, model fusion plays a pivotal role in federated learning (Konečný et al., 2016; McMahan et al., 2017; Wang et al., 2020) when the local updates are collected to make a global update. FedAvg (McMahan et al., 2017) is a classical merging algorithm that directly computes the average of the local models as the updated global model. Recent studies propose to incorporate the permutation symmetry to align the neurons of different end models (Wang et al., 2020; Singh & Jaggi, 2020; Ainsworth et al., 2023). However, these methods fail to achieve a desirable performance when tackling transformer-based models (Jin et al., 2023).

# 7. Conclusion

In this paper, we introduced rotation symmetry as a novel type of parameter space symmetry for transformers, extending the concept of permutation symmetry to continuous spaces. Building on this foundation, we proposed a theoretically optimal parameter matching algorithm to enhance the fusion of transformer models in a plug-and-play manner. To validate our approach, we conducted extensive experiments on real-world NLP and vision benchmarks. The results demonstrated that incorporating rotation symmetry effectively and efficiently facilitates transformer model fusion, showcasing our method's practical utility.

## Impact Statement

Our study provides novel insights into the parameter space symmetry of transformers and establishes a practical framework for advancing model fusion techniques. Bridging theoretical innovations with practical applications, this work reveals the potential for leveraging parameter space symmetries in deep learning research. There are many potential societal consequences of our work, none of which we feel must be specifically highlighted here.

## Acknowledgment

This work is supported in part by the National Science Foundation (NSF) under grants IIS-2006844, IIS-2144209, IIS-2223769, CNS-2154962, BCS-2228534, and CMMI-2411248; the Office of Naval Research (ONR) under grant N000142412636; and the Commonwealth Cyber Initiative (CCI) under grant VV-1Q24-011.

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

# A. Proof

**Theorem 4.1.** *The following optimization problem has a closed-form solution.*

$$\min_{\boldsymbol{R}_1, \boldsymbol{R}_2 \in \mathcal{R}} \left\| \begin{bmatrix} \boldsymbol{W}_{Q_1}^\top & \boldsymbol{W}_{Q_2}^\top \\ \boldsymbol{b}_{Q_1} & \boldsymbol{b}_{Q_2} \end{bmatrix} \begin{bmatrix} \boldsymbol{R}_1 \\ -\boldsymbol{R}_2 \end{bmatrix} \right\|_F^2 + \left\| \begin{bmatrix} \boldsymbol{W}_{K_1}^\top & \boldsymbol{W}_{K_2}^\top \\ \boldsymbol{b}_{K_1} & \boldsymbol{b}_{K_2} \end{bmatrix} \begin{bmatrix} \boldsymbol{R}_1 \\ -\boldsymbol{R}_2 \end{bmatrix} \right\|_F^2. \tag{13}$$

*The solution is given by*

$$\boldsymbol{R}_1 = \boldsymbol{U}\boldsymbol{V}^\top, \boldsymbol{R}_2 = \boldsymbol{I}, \tag{14}$$

*where $\boldsymbol{I}$ is the identity matrix and $\boldsymbol{U}\boldsymbol{\Sigma}\boldsymbol{V}^\top = \boldsymbol{W}_{Q_1}\boldsymbol{W}_{Q_2}^\top + \boldsymbol{W}_{K_1}\boldsymbol{W}_{K_2}^\top + \boldsymbol{b}_{Q_1}^\top \boldsymbol{b}_{Q_2} + \boldsymbol{b}_{K_1}^\top \boldsymbol{b}_{K_2}$ is the result of eigendecomposition.*

*Proof.* We first show the process of converting the optimization problem to an Orthogonal Procrustes problem (Schönemann, 1966; Gower & Dijksterhuis, 2004). We can obtain that the optimization problem is equivalent to the following one.

$$\min_{\boldsymbol{R}_1, \boldsymbol{R}_2 \in \mathcal{R}} \left\| \boldsymbol{W}_{Q_1}^\top \boldsymbol{R}_1 - \boldsymbol{W}_{Q_2}^\top \boldsymbol{R}_2 \right\|_F^2 + \left\| \boldsymbol{b}_{Q_1} \boldsymbol{R}_1 - \boldsymbol{b}_{Q_2} \boldsymbol{R}_2 \right\|_F^2 + \left\| \boldsymbol{W}_{K_1}^\top \boldsymbol{R}_1 - \boldsymbol{W}_{K_2}^\top \boldsymbol{R}_2 \right\|_F^2 + \left\| \boldsymbol{b}_{K_1} \boldsymbol{R}_1 - \boldsymbol{b}_{K_2} \boldsymbol{R}_2 \right\|_F^2. \tag{15}$$

Considering that $\boldsymbol{R}_1, \boldsymbol{R}_2 \in \mathcal{R}$ and $\boldsymbol{R}_1, \boldsymbol{R}_2$ are nonsingular matrices, we let $\boldsymbol{R} = \boldsymbol{R}_1 \boldsymbol{R}_2^{-1}$ and convert Equation (15) to the following one:

$$\min_{\boldsymbol{R}, \boldsymbol{R}_2 \in \mathcal{R}} \left\| \left( \boldsymbol{W}_{Q_1}^\top \boldsymbol{R} - \boldsymbol{W}_{Q_2}^\top \right) \boldsymbol{R}_2 \right\|_F^2 + \left\| \left( \boldsymbol{b}_{Q_1} \boldsymbol{R} - \boldsymbol{b}_{Q_2} \right) \boldsymbol{R}_2 \right\|_F^2 + \left\| \left( \boldsymbol{W}_{K_1}^\top \boldsymbol{R} - \boldsymbol{W}_{K_2}^\top \right) \boldsymbol{R}_2 \right\|_F^2 + \left\| \left( \boldsymbol{b}_{K_1} \boldsymbol{R} - \boldsymbol{b}_{K_2} \right) \boldsymbol{R}_2 \right\|_F^2. \tag{16}$$

As multiplying $\boldsymbol{R}_2$ preserves the Frobenius norm, we can remove the $\boldsymbol{R}_2$ terms from the objective and obtain an Orthogonal Procrustes problem as follows:

$$\min_{\boldsymbol{R} \in \mathcal{R}} \left\| \boldsymbol{W}_{Q_1}^\top \boldsymbol{R} - \boldsymbol{W}_{Q_2}^\top \right\|_F^2 + \left\| \boldsymbol{b}_{Q_1} \boldsymbol{R} - \boldsymbol{b}_{Q_2} \right\|_F^2 + \left\| \boldsymbol{W}_{K_1}^\top \boldsymbol{R} - \boldsymbol{W}_{K_2}^\top \right\|_F^2 + \left\| \boldsymbol{b}_{K_1} \boldsymbol{R} - \boldsymbol{b}_{K_2} \right\|_F^2. \tag{17}$$

We take a look at the first term $\left\| \boldsymbol{W}_{Q_1}^\top \boldsymbol{R} - \boldsymbol{W}_{Q_2}^\top \right\|_F^2$ and have:

$$\begin{aligned}
&\min_{\boldsymbol{R} \in \mathcal{R}} \left\| \boldsymbol{W}_{Q_1}^\top \boldsymbol{R} - \boldsymbol{W}_{Q_2}^\top \right\|_F^2 \\
&= \min_{\boldsymbol{R} \in \mathcal{R}} \left\langle \boldsymbol{W}_{Q_1}^\top \boldsymbol{R} - \boldsymbol{W}_{Q_2}^\top, \boldsymbol{W}_{Q_1}^\top \boldsymbol{R} - \boldsymbol{W}_{Q_2}^\top \right\rangle_F \\
&= \max_{\boldsymbol{R} \in \mathcal{R}} \left\langle \boldsymbol{W}_{Q_1}^\top \boldsymbol{R}, \boldsymbol{W}_{Q_2}^\top \right\rangle_F \\
&= \max_{\boldsymbol{R} \in \mathcal{R}} \mathrm{tr} \left( \boldsymbol{R}^\top \boldsymbol{W}_{Q_1} \boldsymbol{W}_{Q_2}^\top \right) \\
&= \max_{\boldsymbol{R} \in \mathcal{R}} \left\langle \boldsymbol{R}, \boldsymbol{W}_{Q_1} \boldsymbol{W}_{Q_2}^\top \right\rangle_F.
\end{aligned} \tag{18}$$

Similarly, we can convert Equation (17) to the following one:

$$\max_{\boldsymbol{R} \in \mathcal{R}} \left\langle \boldsymbol{R}, \boldsymbol{W}_{Q_1} \boldsymbol{W}_{Q_2}^\top + \boldsymbol{W}_{K_1} \boldsymbol{W}_{K_2}^\top + \boldsymbol{b}_{Q_1}^\top \boldsymbol{b}_{Q_2} + \boldsymbol{b}_{K_1}^\top \boldsymbol{b}_{K_2} \right\rangle_F. \tag{19}$$

We then conduct the singular value decomposition to the matrix $\boldsymbol{W}_{Q_1} \boldsymbol{W}_{Q_2}^\top + \boldsymbol{W}_{K_1} \boldsymbol{W}_{K_2}^\top + \boldsymbol{b}_{Q_1}^\top \boldsymbol{b}_{Q_2} + \boldsymbol{b}_{K_1}^\top \boldsymbol{b}_{K_2} = \boldsymbol{U}\boldsymbol{\Sigma}\boldsymbol{V}^\top$ and have:

$$\begin{aligned}
&\max_{\boldsymbol{R} \in \mathcal{R}} \left\langle \boldsymbol{R}, \boldsymbol{W}_{Q_1} \boldsymbol{W}_{Q_2}^\top + \boldsymbol{W}_{K_1} \boldsymbol{W}_{K_2}^\top + \boldsymbol{b}_{Q_1}^\top \boldsymbol{b}_{Q_2} + \boldsymbol{b}_{K_1}^\top \boldsymbol{b}_{K_2} \right\rangle_F \\
&= \max_{\boldsymbol{R} \in \mathcal{R}} \mathrm{tr} \left( \boldsymbol{V}\boldsymbol{\Sigma}\boldsymbol{U}^\top \boldsymbol{R} \right) \\
&= \max_{\boldsymbol{R} \in \mathcal{R}} \mathrm{tr} \left( \boldsymbol{\Sigma}\boldsymbol{U}^\top \boldsymbol{R}\boldsymbol{V} \right) \\
&= \mathrm{tr}(\boldsymbol{\Sigma}),
\end{aligned} \tag{20}$$

where $\boldsymbol{U}^\top \boldsymbol{R}\boldsymbol{V} = \boldsymbol{I}$, *i.e.*, $\boldsymbol{R} = \boldsymbol{U}\boldsymbol{V}^\top$. Finally, we incorporate $\boldsymbol{R} = \boldsymbol{U}\boldsymbol{V}^\top$ into $\boldsymbol{R} = \boldsymbol{R}_1 \boldsymbol{R}_2^{-1}$ and let $\boldsymbol{R}_2 = \boldsymbol{I}$ for simplicity. Subsequently, we have $\boldsymbol{R}_1 = \boldsymbol{U}\boldsymbol{V}^\top$ and $\boldsymbol{R}_2 = \boldsymbol{I}$ as a closed-form solution of the optimization problem in Theorem 4.1.

□

From the proof, we can observe that the optimization problem in Theorem 4.1 has infinite pairs of solutions $\{\boldsymbol{R}_1, \boldsymbol{R}_2\}$ regarding the value of $\boldsymbol{R}_2$. In this paper, we let $\boldsymbol{R}_2 = \boldsymbol{I}$ for simplicity, which makes model 2 an anchor model. However, the value of $\boldsymbol{R}_2$ can be further adjusted to benefit some data-dependent model fusion techniques (Singh & Jaggi, 2020; Jin et al., 2023). We leave this part to future work.

## B. Comparison with Related Work

Git Re-Basin (Ainsworth et al., 2023) introduces three model fusion algorithms: weight matching, activation matching, and straight-through matching, based on permutation symmetry. While these methods are effective for CNNs and MLPs, they do not easily extend to transformers due to the discrete nature of permutation symmetry. In a discrete space, the parameter matching problem is equivalent to solving a sum of bilinear assignment problems, which is NP-hard (Ainsworth et al., 2023). To address this limitation, we introduce rotation symmetry, which extends the symmetry space to a continuous domain, allowing for a closed-form solution to parameter matching. Unlike permutation symmetry, rotation symmetry provides a more flexible and efficient solution for aligning parameters in transformers.

Additionally, our study emphasizes the advantage of weight matching as a plug-and-play module for model fusion, as it preserves model functionality while reducing the inner distance of end models after matching. OT-ACTS (Imfeld et al., 2024) also includes a parameter matching step, leveraging optimal transport for model fusion. However, its matching module is restricted to permutation operations, and its fusion method is limited to simple merging. In contrast, our approach generalizes parameter matching to rotation operations and integrates it with more advanced merging strategies, demonstrating the potential of parameter matching to enhance model fusion.

A concurrent work by Tran et al. (2024) explores a similar form of rotation symmetry. However, their focus is on constructing functionally equivalent networks for transformers, whereas we focus on leveraging rotation symmetry for model fusion, proposing a theoretically optimal parameter matching algorithm specifically designed to improve fusion performance.

## C. Experimental Details

**Datasets.** Similar with (Jin et al., 2023), We employ emotion classification and named entity recognition (NER) as the tasks for the main experiments. The emotion classification datasets are extracted from (Bostan & Klinger, 2018). Five of them are selected as in-domain datasets (emoint, ssec, electoraltweets, grounded_emotions, affectivetext), and five others are good datasets (dailydialog, crowdflower, tec, tales-emotion, isear). For the NER task, we choose OntoNotes (Hovy et al., 2006) for model finetuning and CoNLL (Tjong Kim Sang & De Meulder, 2003) for out-of-domain evaluation. Additionally, in the ablation study and subset layers matching, we employ datasets (STS-B, SST-2, CoLA) from GLUE (Wang et al., 2019) to analyze the module performance.

**Baselines.** We compared the performance of our method with two matching baselines:

- **OTFusion** (Singh & Jaggi, 2020) mitigates the positional mismatch of functionally analogous neurons across end models by employing optimal transport to align neurons layer-wise, thus enabling more effective model merging.
- **Git Re-Basin** (Ainsworth et al., 2023) matches the parameters of end models through learned permutation symmetries in the parameter space, aiming to transform their parameters into a shared basin within the loss landscape and facilitate seamless merging.

Meanwhile, to address RQ1, we further select six merging methods as baselines:

- **Simple** (Wortsman et al., 2022) directly averages the weight matrices of independently pretrained end models.
- **Fisher** (Matena & Raffel, 2022) approximates each end model's posterior as a Gaussian distribution with a precision matrix derived from its Fisher information, thereby casting model merging as a Maximum Likelihood Estimation (MLE) problem over these posterior distributions.
- **RegMean** (Jin et al., 2023) formulates model merging as a regression problem, aiming to minimize the Euclidean distance between the predictions of the merged model and individual end models over a set of input samples.
- **OT-WTS** (Imfeld et al., 2024) extends OTFusion by introducing soft neuron alignment, which replaces the rigid one-to-one neuron matching with a more flexible, weighted correspondence between neurons. Subsequent model merging is then guided by these soft alignment weights.
- **OT-ACTS** and **OT-ACTS-EMD** (Imfeld et al., 2024) are two variants of **OT-WTS** that match neurons based on their

activations rather than directly aligning parameter matrices. Specifically, OT-ACTS measures activation similarity using Euclidean distance, while OT-ACTS-EMD employs Earth Mover's Distance for a more fine-grained comparison.

The code implementation is based on (Jin et al., 2023)[2] for NLP tasks and (Imfeld et al., 2024)[3] for vision tasks.

## D. Limitations

In this study, we proposed a theoretically optimal parameter matching algorithm based on rotation symmetry. However, the optimality guarantees of our method are subject to several important prerequisites.

First, our theoretical analysis in Theorem 4.1 focuses on the binary model matching case, where the optimal solution is independent of the choice of anchor model. This property, however, does not naturally extend to the multi-model setting. When matching multiple models, a straightforward extension involves selecting a single model as an anchor and aligning the others to it using pairwise rotation-based matching. Yet, in this case, the global optimality depends on the definition of the overall distance metric and may not be preserved. An alternative is iterative pairwise matching, but this introduces path dependency, i.e., the final result can vary depending on the order of merging. Extending our framework to the optimal multi-model matching problem remains an open challenge for future work.

Second, the scope of our parameter matching algorithm is within the rotation symmetry, as defined in Equation (5). While this serves as a tractable and well-motivated instance of parameter space symmetry, we recognize that Transformers exhibit a richer set of structural symmetries beyond permutation, rotation, and rescaling. For example, recent work suggests that geometric properties of normalization and softmax layers may enlarge the symmetry set of previous model layers (Kunin et al., 2021; Lim et al., 2024a). Characterizing the complete symmetry group of transformers remains an open question.

We also considered the possibility of extending rotation matrices to invertible matrices in terms of parameter space symmetries. Although invertible matrices expand the symmetry set, they are less suitable for practical parameter matching. Specifically, our closed-form solution in Equation (9) relies on the orthogonality of the rotation matrices. If replacing them with general invertible matrices, it becomes difficult to derive a closed-form solution for Equation (9), leading to suboptimal solution and increased complexity. Additionally, parameter matching requires computation of the inverse matrix of every $\boldsymbol{R}$, which is impractical for invertible matrices. In contrast, orthogonal matrices benefit from computationally efficient inverses (via transposition) and theoretical simplicity.

Finally, we apply the rescaling symmetry after optimizing for rotation-based matching. However, our sequential approach (first optimizing rotation $\boldsymbol{R}$, then rescaling weight $\alpha$) does not guarantee global optimality for the joint optimization over $(\boldsymbol{R}, \alpha)$. Deriving a globally optimal solution would require solving Equation (12) in closed form, which remains non-trivial. Hence, we consider our method as a practical approximation to the joint optimization solution which balances theoretical soundness with computational efficiency. Exploring a fully joint optimization framework for rotation and rescaling symmetries can be a promising direction for future research.

---

[2]https://github.com/bloomberg/dataless-model-merging
[3]https://github.com/graldij/transformer-fusion

