# OpenReview forum: "Beyond the Permutation Symmetry of Transformers: The Role of Rotation for Model Fusion"
_ICML.cc/2025/Conference — ICML 2025 spotlightposter_

### Official Review · Reviewer_RtZb · 2025-03-12

**Overall Recommendation:** 3

**Summary:**

In this paper, the authors identify a neural network (NN) parameter symmetry beyond the well-studied permutation symmetry. In particular, they show that the weights of self-attention layers are governed by *rotation symmetry*, i.e. one can transform the query, key, value, output matrices by appropriate rotation matrices/their transposes and maintain the function computed by the attention mechanism intact. They proceed by leveraging this symmetry to improve model fusion for self-attention layers. They devise a new parameter matching algorithm by showing that the optimisation problem of aligning self-attention weights up to rotation admits a closed form solution that requires solving an eigendecomposition problem. They additionally enhance their matching algorithm by another step that accounts for weight scaling symmetries, while in the case of a Transformer, the MLPs are aligned using the parameter matching algorithm of Ainsworth et al., 2023. The method is experimentally tested on fusing language and vision transformers, showcasing improved downstream task performance compared to simple (averaging) fusion and other baselines.

**Claims And Evidence:**

- The main claim of the paper is that taking rotation symmetry into account when aligning self-attention weights can improve the downstream performance of fused Transformer models. Although, experimentally this generally seems to be true, I think that the authors have overclaimed in their abstract by mentioning that their "matching algorithm substantially improves model fusion", as this is not clear from the experimental results (Tables 1 and 2). In specific, in several cases the matching algorithm seems to be on par or slightly improves simple fusion, therefore the word "substantially" does not seem to follow from the results.

- Additionally, the authors mention multiple times that they "introduce rotation symmetry" (e.g. Contribution 1), which I believe is misleading, as it has been discussed before in the work of Tran et al., 2024 for weight space learning (neural functionals). I think the phrasing needs to be changed in the text to clarify that rotation symmetry is not a contribution, rather it is taken advantage of to improve model fusion.

**Essential References Not Discussed:**

Regarding permutation symmetry matching, I believe that the paper misses two recent algorithms that improve upon Ainsworth et al.,2023:
- Peña et al., Re-basin via implicit Sinkhorn differentiation, CVPR'23
- Navon et al., Equivariant Deep Weight Space Alignment, ICML'24.

Additionally, regarding weight space symmetries, I believe that the authors should have dedicated more space in mentioning the ongoing efforts on neural functionals/metanetworks (e.g. Navon et al., ICML'23, Zhou et al., NeurIPS'23, Lim et al., ICLR'24 etc.), while the have missed two important references on scaling symmetries (which is something that is taken into account in the current work):
- Kalogeropoulos et al., Scale Equivariant Graph Metanetworks, NeurIPS'25
- Godfrey et al., NeurIPS'22: this is mentioned, but not in the paragraph concerning scaling symmetries.

**Experimental Designs Or Analyses:**

- **Weakness**. The experimental section provides evidence, at least partial, to the claims. However, it was unclear to me how several experimental design choices were made. For example,
    - Why did the authors choose these particular Transformers (RoBERTa, DeBERTa, ViT)? How would model fusion behave on larger models? Have the authors considered extending their experimental section with more recent architectures, or architectures from different domains? I do not intend to imply that this is necessary, rather than it should be made clear why those particular choices were made.
  - Similarly, how did the authors choose these particular baselines (apart from the obvious simple averaging one)?
  - Why do the authors match only the self-attention layers and do not perform merging on the classifier? Perhaps an ablation study on that would help.
  - Would it be possible to use a variant of the matching algorithm to match more than two models, akin to model soups? This would further strengthen the impact of this work.
  - How does each component of the overall matching algorithm contribute to the resulting downstream performance (e.g. permutation matching of MLPs compared to rotation matching of self-attention)?

**Methods And Evaluation Criteria:**

- **Practicality**. The proposed methodology for self-attention weight alignment is technically sound and can be easily implemented in practice. Additionally, I appreciated the fact that the authors include in their framework the MLP alignment, as well as the scaling symmetries of self-attention weights, making their approach practical for fusion of real-world Transformers.
- See Experimental Designs Or Analyses for my comments on empirical evaluation.

**Other Comments Or Suggestions:**

-

**Other Strengths And Weaknesses:**

**Strengths**
- *Importance/Significance*. With the growing availability of trained models, fusing the knowledge embedded in their weights is arguably an important problem, as it reduces the need to train new models (potentially bigger and potentially on new datasets). Since Transformers are currently one of the most popular NN technologies, designing an improved and specialised fusion algorithm for them is a crucial step.
- *Novelty*. Although the rotation symmetry of Transformers has already been studied, it has not been examined in the context of model fusion. To the best of my knowledge, the rotation symmetry parameter matching algorithm provided by the authors is novel.

**Weaknesses**
- As previously mentioned, the main weaknesses are that some claims need to be modified and that the experimental choices need to be explained more thoroughly, and potentially extended.

**Questions For Authors:**

Additional questions that I believe need to be discussed in the paper:

1) When the authors take scaling into account, it seems to me that the way it is done, breaks the optimality of the matching algorithm. Could the authors comment on that? Would it be possible to devise a matching algorithm that is optimal w.r.t both scaling and rotation symmetries?
2) Would it be possible to take into account the symmetries of the rest of the transformer components: e.g. LayerNorm and Softmax?
3) Are rotation symmetries the only ones in self-attention layers? Perhaps there exist a larger group of transformations to which the layer is invariant. Could the authors discuss this?

**Relation To Broader Scientific Literature:**

Although I have not followed the entire literature on model merging, the paper is mostly well-contextualised: it mentions naive (non-symmetry) merging algorithms and compares against them, while it also discusses permutation symmetry matching, which is the most well-studied one.

**Theoretical Claims:**

The optimal solution of the matching algorithm, even though seems straightforward to derive, is an important result and strengthens the credibility of the algorithm (recall that MLP permutation matching algorithms do not admit optimal closed-form solutions beyond 2 layers). The proof is concise and seems correct.

---

> ### Author Rebuttal · Authors · 2025-04-01
>
> Thanks for your constructive feedback. We will include additional results and discussions in the revision. We believe that our paper will be much stronger thanks to your efforts.
>
> **Response to claims**: We thank the reviewer for the suggestion, and we will adjust the wording in the next version of our manuscript. Additionally, we would like to clarify that we have included [1] as a concurrent work in the discussion.
>
> **Response to W1**:
> We thank the reviewer for this suggestion. Our backbone model selection follows previous studies on model fusion [2,3]. Specifically, we deliberately selected models representing different transformer architectures.
>
> While experiments on larger models would indeed be valuable, they would require substantially more computing resources.
> Due to computing resource constraints, we focused on models that could be effectively studied within our available infrastructure.
>
> **Response to W2**:
> For clarity, the three model fusion methods (Fisher, Regmean, and OT) are not baselines in our evaluation, but rather backbone methods that we enhance with our parameter matching algorithm.
> To the best of our knowledge, our work represents the first parameter matching algorithm specifically designed to enhance model fusion performance
> We specifically selected these methods because they achieve desirable performance and provide a diverse set of merging strategies to demonstrate the general applicability of our matching algorithm.
>
> **Response to W3**:
> We first would like to clarify that we follow previous studies [2] to leave classification heads unmerged.
> Unlike attention layers that capture generalizable patterns, classifier layer parameters are highly task-specific in nature.
> Even minor modifications to downstream tasks (e.g., altering label orders in classification tasks or targeting different output distributions) result in entirely different optimal parameter values for classifier heads.
> This task-specificity makes merging classifier heads conceptually unsound without task alignment information.
>
> **Response to W4**:
> Please refer to our response to reviewer **ydmT**.
>
> **Response to W5**:
> Thank you for this insightful question that helps clarify component contributions. We conducted an ablation study isolating the effects of matching different components:
>
> |  | Fisher | Regmean | OT |
> |  --- | -------- | -------- | -------- |
> | FFN-only | 12.21     | 11.89     | 32.08     |
> | Attn-only | **20.21**     | 12.94     | 28.66     |
> | FFN+Attn | 18.61    | **15.31**     | **32.50**     |
>
> For Fisher and RegMean, attention matching provides more contributions to model fusion. For OTFusion, FFN matching provides more contributions. These differences highlight how the underlying fusion method interacts with component matching. We will include the results in our revised manuscript.
>
> **Response to references**: We are committed to adding the suggested references to Related Work of our next revision.
>
> **Response to Q1**:
> We thank the reviewer for this thoughtful question. You raise an important theoretical point about joint optimization of rotation and scaling. It's true that our sequential approach (first optimizing rotation $R$, then scaling $\alpha$) cannot guarantee global optimality for the joint ($R$, $\alpha$) optimization problem. Proving that sequential optimization yields the global optimum would be non-trivial, as calculating the general solution of Eq. (12) without knowing the specific values of matrices is complex.
> We can view our approach (sequentially optimizing $R$ and $\alpha$) as a practical approximation to the joint optimization problem. Developing an algorithm that jointly optimizes for both rotation and scaling symmetries would be an interesting extension to our method, though it would likely come with increased computational complexity. Our current method balances theoretical soundness with computational efficiency.
>
> **Response to Q2**: We believe there might be no symmetries for Softmax and LayerNorm that require matching since these modules do not contain parameter matrices to be rotated.
>
> **Response to Q3**: We thank the reviewer for this thoughtful question. While our work focuses on rotation symmetries, we acknowledge that transformers may exhibit additional symmetries beyond the permutation, rotation, and scaling transformations we address.
> The transformer architecture, with its complex interplay of components, likely possesses a rich symmetry structure that extends beyond what we have explored. Characterizing the complete symmetry group of transformers remains an open question. We identify this as an important direction for future research.
>
> [1] Tran, Viet-Hoang, et al. "Equivariant Neural Functional Networks for Transformers." arXiv 2024.
>
> [2] Jin, Xisen, et al. "Dataless knowledge fusion by merging weights of language models." ICLR 2023
>
> [3] Imfeld, Moritz, et al. "Transformer fusion with optimal transport." ICLR 2024

---

> > ### Comment · Reviewer_RtZb · 2025-04-04
> >
> > I thank the authors for their response.
> >
> > - Quick clarification: softmax and normalization layers do induce symmetries (translation and scaling resp., see Neural Mechanics: Symmetry and Broken Conservation Laws in Deep Learning Dynamics, Kunin et al., ICLR'21) that I would expect to affect the overall symmetry group of the Transformer weights (beyond rotations). In addition, if I am not mistaken the rotation matrices can be extended to arbitrary invertible matrices (general linear group). I think the authors should study those more thoroughly and discuss this observation in their paper - perhaps as a limitation/room for improvement, i.e. that not all symmetries are considered for parameter matching.

---

> > > ### Author Response · Authors · 2025-04-07
> > >
> > > We thank the reviewer for these insightful follow-up comments.
> > > Regarding softmax and normalization layers, we appreciate you bringing this related work [1] to our attention. We agree that the geometric properties of these components indeed enlarge the symmetry set of previous model layers.
> > > For the scope of rotation symmetry, we agree that the orthogonal matrices constraint can be extended to general invertible matrices as in our previous response to reviewer **QTZj**, but it comes with practical challenges in parameter matching, including higher computational complexity and suboptimal solution. Please refer to our response to reviewer **QTZj** for more details.
> > > Both of these points represent important supplements to the scope of parameter-space symmetries in transformers. We commit to adding a separate limitations section to discuss these broader symmetry considerations and corresponding challenges for future studies.
> > >
> > > [1] Kunin, Daniel, et al. "Neural Mechanics: Symmetry and Broken Conservation Laws in Deep Learning Dynamics." ICLR 2021

---

### Official Review · Reviewer_QTZj · 2025-03-12

**Overall Recommendation:** 4

**Summary:**

The paper studies transformer parameter symmetries. Specifically, it explains how to weight space average two attention layers modulo not only permutation symmetries but also rotation symmetries. Experimental results show that considering this extra symmetry leads to better alignment between different trained transformers.

## update after rebuttal

The authors have reasonably addressed my concerns, and so I increased my rating to 4.

**Claims And Evidence:**

Most claims are well supported.

**Essential References Not Discussed:**

N/A

**Experimental Designs Or Analyses:**

The experimental design seems reasonable.

**Methods And Evaluation Criteria:**

The benchmarks seem to align with prior work.

**Other Comments Or Suggestions:**

1. Include the baseline scores in the tables. I.e the scores of the models that are merged.
2. Comment on the fact that CIFAR10 accuracies are extremely low after merging. (30%)

**Other Strengths And Weaknesses:**

1. The paper considers rotation symmetries and also rotation symmetries + scaling. One could also straightforwardly consider invertible linear transformations $A$ in general since $QA A^{-1} K^T = QK^T$.

**Questions For Authors:**

1. Is there any intuition/reason as to why the merged networks do not perform well? E.g. 30% accuracy on CIFAR10 sounds very low.
2. Would it be possible to extend to general invertible linear transformations $A$ as described under "Other Strengths And Weaknesses" above?

**Relation To Broader Scientific Literature:**

The paper considers a further symmetry than prior work. I am not aware of prior work exploring rotation symmetries in attention layers.

**Theoretical Claims:**

I haven't checked the proofs. The results seem plausible.

---

> ### Author Rebuttal · Authors · 2025-04-01
>
> Thank you very much for your thoughtful feedback. We are honored to have this valuable chance to address your raised concerns and questions. We believe that our paper will be much stronger thanks to your efforts.
>
> **Response to comments and Q1**:
> We thank the reviewer for this important question.
> To clarify, our choice of end ViT models strictly follows previous works [1], resulting in similar performance degradation when merging models trained on different tasks (Table 1 in [1]).
> We attribute these low metric values to two key factors:
> 1. Task divergence. The end ViT models were fine-tuned on substantially different tasks, leading to specialized parameters that conflict when naively merged.
> 2. Non-convexity. ViTs exhibit highly non-convex loss landscapes, making naive interpolation between models challenging compared to simpler models such as MLPs.
>
> Actually, merging large models on different tasks remains challenging. This also motivates our parameter matching strategy which improves model merging without access to any training data. We are committed to including the metric values of the end models in the next version of our paper.
>
> **Response to Q2**: We thank the reviewer for providing this insightful comment. We find that all orthogonal matrices in the rotation symmetry can be directly replaced by invertible matrices without any barrier. However, invertible matrices are not applicable for parameter matching. In our parameter matching algorithm, the orthogonal matrix is an important promise for Theorem 1. If using invertible matrices, Eq. (9) can only be solved by gradient descent, losing the guarantee of global optimum. In addition, parameter matching requires computation of the inverse matrix of every $R$, which is impractical for invertible matrices. In contrast, the inverse matrix of the orthogonal matrix is just its transpose. In summary, if we only consider the rotation symmetry for transformers, then yes, the orthogonal matrices in the rotation symmetry can be replaced by invertible matrices without barrier; while if we consider the parameter matching based on rotation symmetry, then no, using invertible matrices results in suboptimal parameter matching and large complexity.
>
> We extend our best gratitude for your efforts in the rebuttal phase. We highly value the opportunity to improve our paper. We will include the additional results and discussions in the next version of our paper. We sincerely appreciate your time and consideration.
>
> [1] Imfeld, Moritz, et al. "Transformer fusion with optimal transport." ICLR 2024

---

> > ### Comment · Reviewer_QTZj · 2025-04-04
> >
> > I thank the authors for their response.
> >
> > > Task divergence. The end ViT models were fine-tuned on substantially different tasks, leading to specialized parameters that conflict when naively merged.
> >
> > Are not both ViT models trained on CIFAR10? That was my interpretation of [1]. Quote from that paper:
> > > First, we train individual models from scratch on each dataset until convergence.
> > We ensure model diversity by initializing each model with different seed values and different batch
> > randomization. This results in unique models with similar performance but located in diverse parts of
> > the landscape, and whose suitable fusion can improve performance.
> >
> > Further, I would like to ask what the corresponding table in that paper is to the experiments in this paper. Finally, is there a reason for not including the best method from that paper? It seems like "OT-ACTS" outperforms "OT-ACTS (EMD)".

---

> > > ### Author Response · Authors · 2025-04-07
> > >
> > > We thank the reviewer for these additional concerns.
> > > Regarding the factors affecting ViT performance: After revisiting the paper [1], we acknowledge our misunderstanding of the experimental settings and agree that the end models were trained on the same dataset with different random seeds and batch selections. In this case, the highly non-convex nature of ViT remains the primary explanation for the low performance of model fusion. Additionally, task divergence remains relevant for explaining challenges in our language model fusion settings.
> > >
> > > Regarding your question about the corresponding table: Our ViT experimental settings align with **Table 1** in [1]. For your question about baseline selections, we selected OT-ACTS (EMD) as our baseline despite OT-ACTS showing better performance because hard alignment guarantees functional equivalence of the matched model, making it a feasible baseline for both model fusion (in Table 2 of our paper) and parameter matching (in Figure 3 of our paper).
> > > To address your concern about including the best methods, we have conducted additional experiments using our parameter matching approach with OT-Fusion variants (including soft alignment):
> > >
> > > |         | OT-ACTS (EMD) | OT-WTS |  OT-ACTS |
> > > | ------- | ---- | -------- |---|
> > > | w/o Match         | 32.08     | 57.11 | 61.15  |
> > > | FFN-only          | 32.08     | 57.11 |  61.15 |
> > > | Attn-only         | 28.66     | 56.16 |  60.07 |
> > > | FFN+Attn          | 32.50     | 57.16 | 61.23 |
> > > | FFN+Attn (scale)  | **32.53** | **57.17** |  **61.25** |
> > >
> > > These results demonstrate that our parameter matching approach consistently improves performance across all model fusion methods. We commit to including these additional results in the next version of our paper.
> > > We sincerely appreciate your careful examination of our work and your efforts in improving our paper.
> > >
> > > [1] Imfeld, Moritz, et al. "Transformer Fusion with Optimal Transport." ICLR 2024

---

### Official Review · Reviewer_sNWf · 2025-03-13

**Overall Recommendation:** 4

**Summary:**

The paper introduces rotation symmetry in transformers, extending permutation symmetry from discrete to continuous spaces. It demonstrates theoretically and empirically that rotating query-key and value-output parameter matrices preserves functional equivalence. The main contribution is a theoretically optimal algorithm for matching parameters during model fusion, significantly enhancing performance across NLP and vision benchmarks.

**Claims And Evidence:**

The main claim is that rotation symmetry improves transformer model fusion by reducing distances between parameter sets. This is convincingly supported by extensive empirical evidence showing consistent improvement across multiple fusion methods, transformer architectures, and tasks.

**Essential References Not Discussed:**

All related works are cited to my knowledge.

**Experimental Designs Or Analyses:**

Experimental designs and analyses appear sound and valid. Model fusion benchmarks, comparisons against well-established methods, and the analysis of distances in parameter space are well-executed.

**Methods And Evaluation Criteria:**

The methods and evaluation criteria (real-world NLP and vision tasks, including Emotion, NER, GLUE benchmarks, CIFAR-10) are appropriate, realistic, and convincingly aligned with demonstrating the advantages of their proposed method.

**Other Comments Or Suggestions:**

N/A

**Other Strengths And Weaknesses:**

The introduction of continuous rotation symmetry is original, theoretically insightful, and practically impactful. However, the potential computational overhead in larger transformer models or extremely high-dimensional settings might need further exploration.

**Questions For Authors:**

The correlation between reduced Euclidean distance in parameter space and improved model fusion performance is assumed to be direct. Are there references that provide a theoretical backing for this correlation?

**Relation To Broader Scientific Literature:**

This work substantially builds upon existing literature on permutation symmetry (e.g., Ainsworth et al., 2023; Entezari et al., 2022) and extends it significantly into continuous rotation symmetry for transformers, providing both theoretical novelty and practical utility. It addresses a clear gap regarding transformer-specific symmetries.

**Theoretical Claims:**

The main theorem (Theorem 4.1) provides a closed-form solution for the rotation symmetry optimization problem. The proof is sound and well-structured.

While the derivations in the paper are mathematically elegant, there are concerns regarding their reliance on idealized conditions. In particular, the closed-form solution (Theorem 4.1) leverages perfect orthogonality and well-conditioned eigendecompositions - assumptions that may not hold in high-dimensional, noisy parameter spaces encountered during practical training.

---

> ### Author Rebuttal · Authors · 2025-04-01
>
> Thank you very much for your thoughtful feedback. We are honored to have this valuable chance to address your raised concerns and questions. We believe that our paper will be much stronger thanks to your efforts.
>
> **Response to theoretical claims**:
> Thank you for this thoughtful point. We would like to clarify that the perfect orthogonality isn't an assumption but rather a property that holds for any SVD decomposition.
> Even in noisy parameter spaces, SVD always provides orthogonal singular vectors. Any parameter matrix obtained through practical training, e.g., standard SGD, differentially private SGD, and other optimization methods, can be decomposed this way.
>
> **Response to questions**: Yes, the correlation between reduced Euclidean distance and improved model fusion performance is backed by previous studies [1]. Their theoretical results show that strong convexity and closer end models can boost the utility of direct model fusion. Additionally, experimental results in [1] and our paper support the correlation from an empirical perspective.
>
> We extend our best gratitude for your efforts in the rebuttal phase. We highly value the opportunity to improve our paper. We will include the additional results and discussions in the next version of our paper. We sincerely appreciate your time and consideration.
>
> [1] Wortsman, Mitchell, et al. "Model soups: averaging weights of multiple fine-tuned models improves accuracy without increasing inference time." ICML 2022.

---

### Official Review · Reviewer_ydmT · 2025-03-14

**Overall Recommendation:** 4

**Summary:**

The paper extends the concept of permutation symmetry in MLPs to rotation symmetry for the self-attention layer in the transformers. The authors show that due to the inherent design of self-attention layers, each of the query, key, and value vectors can be rotated without changing the functional representation and thus can be used to match two transformer models. Based on this, authors propose a model merging algorithm for transformer-based models and empirically show that by matching two models, merging can be improved.

**Claims And Evidence:**

Most of the claims made in the paper are well-supported; however, the authors suggest that the proposed matching algorithm can be used to merge multiple models, though they only experiment with merging with two models. Similar work for CNNs suggests that it may not be possible to merge more than two models simultaneously [1].

[1]. Sharma et al., Simultaneous Linear Connectivity of Neural Networks Modulo Permutation

**Essential References Not Discussed:**

Related works are cited and discussed.

**Experimental Designs Or Analyses:**

1. I think authors should show that Linear Mode Connectivity improves after the merging.

2. Moreover, it is not clear to me if the fine-tuned models will move out of the loss basin of the original pre-trained models. This could also explain why matching only the first few layers works, as observed by the authors.

**Methods And Evaluation Criteria:**

Yes, methods is evaluated correctly with the relevant benchmark datasets.

**Other Comments Or Suggestions:**

Line 226
>and can be solved precisely by Hungarian Algorithm (Martello & Toth, 1987).

I believe weight matching and activation matching give an approximate solution; I would refrain from using the word precisely.

**Other Strengths And Weaknesses:**

### Strengths:
1. The paper is well-written and easy to follow.

2. The paper studies an important problem of symmetry for transformer-based models and successfully extends the previous work to transformer models.

3. The closed-loop form for obtaining $R$ is interesting.

4. Experiments are well-designed, and empirical results demonstrate the efficacy of the method.

### Weaknesses:

1. The models are fine-tuned so that may already be in the same basin. As suggested earlier, study the LMC before and after the merging.

2. Experiments on merging more than two models need to be included, or the claim needs to be readjusted otherwise.

**Questions For Authors:**

Why do you think fine-tuning the models will move them outside the loss basin of the original model? Previous work suggests they remain in the same basin [2].

[2]. Model soups: averaging weights of multiple fine-tuned models improves accuracy without increasing inference time

**Relation To Broader Scientific Literature:**

The paper introduces a new concept of symmetry --- rotational symmetry --- which could match transformers. Previous work has only looked into the permutation symmetry for MLP/CNNs, which limited its application to transformers. However, rotational symmetry is more general symmetry, which can be used with transformers. The authors theoretically explain and show how to obtain this rotation symmetry for transformers; this could spur future research.

**Theoretical Claims:**

Yes, the theoretical claims are correct and easy to understand.

---

> ### Author Rebuttal · Authors · 2025-04-01
>
> Thank you very much for your thoughtful feedback. We are honored to have this valuable chance to address your raised concerns and questions. We believe that our paper will be much stronger thanks to your efforts.
>
> **Response to claims and W2**:
> We thank the reviewer for this insightful suggestion.
> In the binary case, the optimality of parameter matching does not rely on the choice of anchor models. However, this might not hold for multiple models.
> When merging multiple end models, a naive extension of our method is to select a single model as an anchor and align all others to it via rotation symmetry, as done in pairwise merging.
> However, the optimality of this approach depends on how the overall distance measure is defined.
> An alternative approach could involve iterative pairwise merging, but this introduces path dependency issues where the final result depends on the order of merging.
> We agree that finding optimal alignments across multiple models is an interesting and important direction and plan to explore it in the future work.
> We will make corresponding adjustments to the claims in the revision of our paper.
>
> **Response to experiments 1**:
> Thank you for this valuable suggestion. We've computed the LMC curves of the ViT model with and without our proposed matching technique on the CIFAR-10 dataset. Our analysis reveals two key observations:
>
> 1. For most interpolation coefficients $\lambda\in[0,1]$, models merged with our matching approach exhibit lower loss values than unmatched models, indicating improved connectivity.
>
> 2. We observe a loss barrier from the LMC curve between the end models used in our experiments. We explain this because ViT exhibits a strong non-convexity [1]. Similar results are recorded in previous works [2].
>
> We will include detailed LMC results in our revised manuscript to fully illustrate these findings.
>
> **Response to experiments 2 and W1**:
> We thank the reviewer for this insightful question. We've analyzed the LMC curves between end models fine-tuned on different NLP tasks and observed that all model pairs (in our settings) exhibit positive loss barriers, confirming that fine-tuning often does move models into different loss basins. However, the magnitude of these barriers varies across tasks.
>
> Importantly, we observe positive loss barriers between the non-anchor original model and its matched counterpart. This provides evidence that our matching technique achieves model rebasin through rotation symmetry.
>
> Regarding why matching only the first few layers is often sufficient: this suggests that rotational symmetry divergence primarily occurs in early layers during fine-tuning, while later layers maintain more consistent representations. This aligns with observations that early layers capture more task-specific features [3].
>
> We acknowledge that precisely characterizing loss basin distributions remains an open problem due to the highly non-convex nature of transformer loss landscapes. Even for simpler architectures like MLPs, loss barrier analysis remains challenging. We believe this is a valuable and insightful direction for future research.
>
> **Response to comments**:
> We believe there might be misunderstanding regarding this sentence. In our manuscript, we mentioned that the **linear assignment problem** can be solved precisely by Hungarian Algorithm, rather than weight matching or activation matching problems.
>
> **Response to questions**:
> We thank the reviewer for this important point. To our knowledge, our experimental setting differs from the model soup paper [4].
> The model soup paper primarily studies merging models trained on **the same dataset** with different hyperparameters, where models likely remain in the same loss basin as they optimize for the same objective.
> In contrast, our experiments merge models fine-tuned on different datasets. Our LMC analysis confirms these models occupy different loss basins, as evidenced by positive loss barriers.
>
> We extend our best gratitude for your efforts in the rebuttal phase. We highly value the opportunity to improve our paper. We will include the additional results and discussions in the next version of our paper. We sincerely appreciate your time and consideration.
>
> [1] Park, Namuk, and Songkuk Kim. "How Do Vision Transformers Work?." ICLR 2022.
>
> [2] Imfeld, Moritz, et al. "Transformer fusion with optimal transport." ICLR 2024
>
> [3] Tenney, Ian, Dipanjan Das, and Ellie Pavlick. "BERT Rediscovers the Classical NLP Pipeline." ACL 2019.
>
> [4] Wortsman, Mitchell, et al. "Model soups: averaging weights of multiple fine-tuned models improves accuracy without increasing inference time." ICML 2022.

---

> > ### Comment · Reviewer_ydmT · 2025-04-02
> >
> > Thank you for the detailed response and for running additional experiments.
> >
> > **Analysis of LMC**
> >
> > Please add a detailed analysis of LMC for different tasks in the final version. LMC gives a better idea of about the effectiveness of matching.
> >
> > I really enjoyed reading the paper; matching transformers with rotational symmetry could be useful for many applications, and a lot of work done on MLP/ConvNets with permutation matching can be extended to transformers with rotational symmetry. I recommend accepting the paper!
> >
> > Edit: Updated the score!

---

> > > ### Author Response · Authors · 2025-04-03
> > >
> > > We are delighted to learn that our rebuttal addressed your concerns. We commit to adding a detailed analysis of the LMC results for different tasks in the final version, including all discussions during the rebuttal phase.
> > >
> > > In light of your further feedback, we respectfully hope you can consider updating the overall recommendation of our paper. Thank you again for your constructive feedback and dedicated efforts in improving our paper.
> > >
> > > Best,
> > >
> > > Authors

---

### Decision · Program_Chairs · 2025-05-01

**Decision:**

Accept (spotlight poster)

**Comment:**

Leveraging rotation symmetry in self-attention layers, the authors propose a parameter matching algorithm that achieves improved empirical performance across diverse ML tasks.

The reviewers recognized the introduction of rotation symmetry in transformer model fusion as a novel, theoretically sound, and practically valuable contribution. Concerns initially raised about potentially overstated empirical claims and the need for a more comprehensive experimental evaluation were largely resolved through discussions with the authors. Overall, the paper was viewed positively for its conceptual originality and prospective impact.

The authors are encouraged to incorporate the important feedback given by the knowledgeable reviewers.